Comparative genomics analysis of the MYB gene family in barley: preliminary insights into evolution and biological function in Blue Qingke

Li Hongyan 1 2 3 4
Yao Youhua 1 2 3 4
Li Xin 1 2 3 4
Cui Yongmei 1 2 3 4
An Likun 1 2 3 4
Ding Baojun 1 2 3 4
Yao Xiaohua 1 2 3 4 yaoxiaohua009@126.com
Wu Kunlun 1 2 3 4 wklqaaf@163.com
1 Academy of Agricultural and Forestry Sciences, Qinghai University, Qinghai , China
2 Qinghai Key Laboratory of Hulless Barley Genetics and Breeding , Academy of Agricultural and Forestry Sciences, Qinghai University, Qinghai , China
3 Qinghai Subcenter of National Hulless Barley Improvement , Academy of Agricultural and Forestry Sciences, Qinghai University, Qinghai , China
4 Laboratory for Research and Utilization of Qinghai Tibet Plateau Germplasm Resources , Academy of Agricultural and Forestry Sciences, Qinghai University, Qinghai , China
Nunes-da-Fonseca Rodrigo
Electronic publication date: 2024 Dec 2
Publication date: 2024
Volume: 12
Electronic Location ID: e18443
Received 2024 Jul 11; Accepted 2024 Oct 11
Copyright: © 2024 Li et al.
Copyright year: 2024
Copyright holder: Li et al.
License: This is an open access article distributed under the terms of the Creative Commons Attribution License, which permits unrestricted use, distribution, reproduction and adaptation in any medium and for any purpose provided that it is properly attributed. For attribution, the original author(s), title, publication source (PeerJ) and either DOI or URL of the article must be cited.
License URL: https://creativecommons.org/licenses/by/4.0/

Keywords: Qingke, MYB family, Anthocyanin, Gene expression

Funding: Natural Science Foundation Project of Qinghai Province 2021-ZJ-950Q National Key Research and Development Project 2022YFD2301300 Agriculture Research System of China CARS-05 This research was supported by the Natural Science Foundation Project of Qinghai Province (2021-ZJ-950Q), the National Key Research and Development Project (2022YFD2301300), and the Agriculture Research System of China (CARS-05). The funders had no role in study design, data collection and analysis, decision to publish, or preparation of the manuscript.

==============================
Background

The Myeloblastosis related (MYB) family is one of the most widely distributed transcription factor families in plants and plays a significant role in plant growth and development, hormone signal transduction, and stress response. There are many reports on MYB family species, but the research on Qingke is still limited.

Methods

This study used comparative genomics methods to analyze gene and protein structure, protein physicochemical properties, chromosome localization, and evolution. A bioinformatics approach was used to systematically analyze the HvMYB gene family. At the milk stage, soft dough stage, and mature stage, White and Blue Qingke grains were selected for RNA sequencing (RNA-seq), among which two proteins interacted (HvMYB and HvMYC). The expression of this gene family was analyzed through RNA-seq, and the expression levels of HvMYB and HvMYC in the grains of two different color varieties were analyzed by quantitative reverse transcription polymerase chain reaction (qRT-PCR). Finally, the interaction between HvMYB and HvMYC was verified by bimolecular fluorescence complementation (BiFC) experiments.

Results

A total of 92 Qingke HvMYB genes were identified and analyzed, and 92 HvMYB proteins were classified into five categories. Cis-acting elements associated with abscisic acid response, light response, and methyl jasmonate (MeJA) response were found in the promoter regions of most MYB genes. Using qRT-PCR combined with RNA-seq analysis showed that MYB gene was highly expressed in the soft dough stage and was varietal specific. Subcellular localization indicated that HvMYB was located in the nucleus and cell membrane, HvMYC was located in the nucleus, cell membrane, and cytoplasm. Through BiFC analysis, it has been proven that HvMYB in the MYB family and HvMYC in the basic helix–loop–helix (bHLH) family can interact. This study provides a preliminary theoretical basis for understanding the function and role of the Qingke MYB gene family and provides a reference for the molecular mechanism of Qingke gene evolution.

Introduction

Qingke (Hordeum vulgare L. var. nudum Hook. f), a variety of barley in the gramineous family that is highly genetically similar to barley (Hordeum vulgare L.), is also called naked barley or hulless barley in the Tibetan Plateau of China, as the lemma and palea are separated from the caryopsis when the grain is ripe (Guo et al., 2020; Yao et al., 2021). Qingke has a nutrient composition structure of “three high and two low” (high protein, fiber, and vitamins and low fat and sugar) and has an extremely high nutritional and dietary value (Guo et al., 2020). The grain color of highland barley is related to the accumulation of different pigments in the aleurone layer, pericarp, and lemma (Jende-Strid, 1993; Strygina, Börner & Khlestkina, 2017), and there are mainly four colors: blue (anthocyanin deposited in the aleurone layer), purple (anthocyanin deposited in the aleurone layer and pericarp), black (pigment deposited in the glume or pericarp), and white (without pigment) (Jia et al., 2016; Suriano et al., 2018; Zeven, 1991). Studies have shown that blue and purple are caused by flavonoid phenolic compounds, and black is caused by plant melanin, oxidation, and polymerization of phenolic compounds (Shoeva et al., 2016). The phenylpropanoid pathway produces lignin, flavonoids, and other metabolites and is regulated by MYB transcription factors (Ma & Constabel, 2019). The blue aleurone trait in barley has only recently evolved, with most wild barley varieties displaying blue aleurone layers, while local and cultivated varieties have blue, purple, and white aleurone layers in various colors (Jia et al., 2020). The blue grain traits of barley are mainly derived from three incomplete dominant Ba genes: Ba1, Ba2, and BaThb (Shen et al., 2013; Singh et al., 2007; Yu et al., 2017; Zheng et al., 2006). In barley, the combination of dominant alleles controlling three transcription factors, MYB, bHLH, and WD40 (MBW complex), regulates anthocyanin synthesis in the aleurone layer (Strygina, Börner & Khlestkina, 2017). Delphinium 3-glucoside is the primary anthocyanin in blue grain barley (Mullick et al., 1958). Previous studies have suggested that anthocyanin accumulation in blue grain barley is positively correlated with the expression levels of the Glutathione S-transferase (HvGST) gene and MbHF35 cluster genes (HvMYB4H, HvMYC4H, and HvF3′5′H). The quantitative trait locus (QTL) associated with the HvGST gene has been preliminarily located at 2.80 Mb (486.18–488.98 Mb) of chromosome 7HL (Jia et al., 2020; Xu et al., 2023).

The MYB family is large and diverse and one of the most widely distributed transcription factor families in plants, being present in all eukaryotes (Ambawat et al., 2013; Dubos et al., 2010). MYB transcription factors have a highly conserved N-terminal MYB DNA-binding domain (DBD), consisting of 1–4 incomplete amino acid repeats (R) and containing about 52 amino acid residues, each encoding three alpha-helices (H1, H2, and H3), of which H2 and H3 form a helix-turn-helix (HTH) structure (Ng, Abeysinghe & Kamali, 2018). MYB transcription factors can bind to target DNA through the HTH structure to regulate the expression of target genes (Rosinski & Atchley, 1998). Of the four MYB proteins, the R2R3-MYB protein is the most abundant type and is specific to plants, responding to both biotic and abiotic stress, as well as primary and secondary metabolism. The R2R3-MYB gene may mainly regulate specific processes in plant evolution during formation (Kranz, Scholz & Weisshaar, 2000; Wilkins et al., 2009). The R2 domain contains a conserved DNA-binding site, and the R3 domain contains a conserved domain that can bind to the bHLH protein (Yin et al., 2021). In plants, the first tryptophan in R3 is replaced by phenylalanine or isoleucine (Ambawat et al., 2013). Studies have found that MYB genes are targets of microRNAs (miRNAs) (Allen et al., 2007). MYB transcription factors are involved in plant development and stress response by binding to MYB cis-elements in target gene promoters, which can regulate the growth of plant roots, stems, leaves, and flowers, and also participate in the biosynthesis of plant secondary metabolites, including anthocyanins, flavonols, and lignin (Lama et al., 2020; Li, Ng & Fan, 2015; Wang, Niu & Zheng, 2021). Numerous studies have shown that MYB transcription factors are widely distributed in plants, are involved in the abscisic acid (ABA) response, and interact with other transcription factors (Ng, Abeysinghe & Kamali, 2018). The first MYB gene identified was v-MYB of avian myeloblastosis virus (AMV) (Klempnauer, Gonda & Bishop, 1982). The first MYB protein discovered was factor C1 in Zea mays (Kranz, Scholz & Weisshaar, 2000). The animal c-MYB gene contains three HTH-encoding repeats (R1R2R3 class genes); thus, it has been suggested that the R2R3-MYB gene is the plant equivalent of c-MYB (Kranz, Scholz & Weisshaar, 2000). Subsequently, MYB transcription factors of numerous diverse species have been identified, most of which activate MYBs and bind to cis-elements in gene promoters to regulate transcription mechanisms and activate gene expression (Ma & Constabel, 2019).

Currently, studies on the MYB gene family have focused on Arabidopsis (Dubos et al., 2010), rice (Katiyar et al., 2012), and maize (Du et al., 2012), while fewer studies have been conducted on the Qingke MYB family. Previous researchers have identified a MYB family gene, HvMYB4H, located at the Blx1 locus on chromosome 4H, which controls the blue aleurone trait in barley (Jia et al., 2020). However, the anthocyanin-related genes that control Qingke’s blue aleurone trait have not been fully elucidated. Moreover, MYB transcription factor is the key factor controlling the anthocyanin phenylpropanoid (Zeng et al., 2020). Therefore, based on published reports of MYB genes in Arabidopsis and rice (Katiyar et al., 2012), firstly, 92 Qingke HvMYB genes were obtained by comparative genomics analysis. Chromosome localization, gene and protein structure, protein physicochemical properties, conserved domains, protein conserved motifs, promoter cis-acting elements, phylogenetic relationships, collinearity with Arabidopsis, rice, and maize were analyzed using bioinformatics methods. Notably, only R2R3-MYB transcription factors have been reported in barley (Tombuloglu et al., 2013), and that the method was one-sided as it was screening R2R3 MYB TFs from barley whole RNA-seq data. Therefore, the identification in this study is crucial to ensure the accuracy and comprehensiveness of the barley MYB gene family. The interaction between HvMYB and HvMYC proteins on cell nucleus and cell membrane was verified by the BiFC method, which laid a foundation for the research and application of HvMYB transcription factors in the grain color of cereal crops.

Materials and methods

Plant materials

The two Qingke varieties ‘White 91-97-3’ (white grain) and ‘Blue Qingke’ (blue grain) were planted in the field with 10 cm plant spacing and 30 cm row spacing, and normal field management was carried out, that is, the watering interval is about 15 days. The grains and leaves of two Qingke varieties are taken from the milk stage, soft dough stage, and mature stage, three biological replicates were performed, frozen in liquid nitrogen and stored at −80 °C, for subsequent gene cloning, RNA-seq, and qRT-PCR analysis. Qingke varieties were collected from the Institute of Crop Breeding and Cultivation, Academy of Agriculture and Forestry Sciences, Qinghai University.

Identification and analysis of MYB family members

The MYB family hmm file was downloaded from the Pfam database (http://pfam-legacy.xfam.org/family/PF00249). The FASTA and GTF data for the barley genome is available at the EnsemblPlants database the barley (https://plants.ensembl.org/Hordeum_vulgare/Info/Index), Arabidopsis (https://plants.ensembl.org/Arabidopsis_thaliana/Info/Index), rice (https://plants.ensembl.org/Oryza_sativa/Info/Index), and maize (https://plants.ensembl.org/Zea_mays/Info/Index) genomes were downloaded from the EnsemblPlants database. Structural domain analysis was performed using SMART (http://smart.embl-heidelberg.de/) and NCBI (https://www.ncbi.nlm.nih.gov/Structure/bwrpsb/bwrpsb.cgi), after removing the repetitive sequences of structural domain names. Only domains co-existing in the NCBI and SMART databases were considered valid domains. The physicochemical properties of HvMYB proteins were analyzed using Expasy Protparma (https://web.expasy.org/protparam/). Subcellular localization was predicted using WoLF PSORT (https://wolfpsort.hgc.jp/). Evolutionary tree analysis was performed using MEGA7 and the neighborhood method (NJ) with 1,000 bootstrap replicates. Covariance analysis was performed using MCscanX software (Threshold: E-value = 1e−10, Num of Blast Hits = 5). The gene structure and chromosome physical location of the HvMYBs in the gene transfer format (GTF) files were extracted. The conserved motifs of the protein gene families were predicted using MEME (http://meme-suite.org/) with a maximum value of 10 for motifs and an optimized motif width of 6–50, and with other parameters set to the system defaults. The results of the Arabidopsis MYB family classification (https://www.arabidopsis.org/results?mainType=general&searchText=MYB&category=genes) were used as the basis for the classification of the Qingke HvMYB family. The cis-acting elements of the HvMYB were analyzed using PlantCARE (http://bioinformatics.psb.ugent.be/webtools/plantcare/html/). The HvMYB gene was identified by comparative genomics and plant transcription factor database (https://planttfdb.gao-lab.org/). The figures were produced using TBtools.

Gene expression analysis in RNA-seq data

The RNA-seq data of Qingke with different grain colors were obtained from the preliminary test in our laboratory, the RNA-seq data has been uploaded to NCBI, and the specific sequence files are in Fig. S1. We retrieved the number of enzyme fragments per kilobase million (FPKM) values for the MYB family at each developmental stage from RNA-seq. The statistical power of this experimental design, calculated in RNASeqPower was 0.37, provided by Novogene Bio (Table S1). The obtained expression data were collated and subjected to analysis using Microsoft Excel 2010 and SPSS 19.0 statistical software.

Candidate gene isolation and expression analysis

White 91-97-3 and Blue Qingke grains were collected at different periods of grain color formation: the milk, soft dough, and mature stages. RNA-seq and qRT-PCR analysis were performed three times on each sample. Identification and analysis of differentially expressed genes with reference to the methodology of Yao et al. (2021).

The total RNA was extracted from the grain of the Qingke according to the instructions for the plant RNA extraction kit (TaKaRa, Beijing, China). The concentration and purity of the RNA were determined using an ultra-micro nucleic acid protein measuring instrument (NanoPhotometer, Munich, Germany), and the quality was measured using 1.0% agarose gel electrophoresis. Reverse transcribed complementary DNA (cDNA) according to the instructions for the cDNA synthesis kit (TaKaRa, Beijing, China) and was stored at −20 °C. Polymerase chain reaction (PCR) amplification was performed using cDNA from qingke leaves as a template (Table S2), and Primer 5.0 was used to design the amplification primers for two genes (Table 1). The PCR amplification and agarose gel electrophoresis methods were used as in a previous study (Li et al., 2024). The target bands were recovered using a column DNA Ge1 Extraction Kit (Sangon Biotech, Shanghai, China), and the target fragments were inserted into the vector of pEasy Blunt (TransGen, Beijing, China) and transformed into E. coli Trans-T1 receptive cells. Three positive clones were selected and sent to Sangon Biotech (Shanghai, China) for sequencing.

Table 1 All primer sequences used in the experiment.

Primer name	Forward primer (5′-3′)	Reverse primer (5′-3′)	Purpose	
HvMYB-1	ATCATCGCTGGCAGGTTG	TCGTGGCGGAATGTGGT	qRT-PCR	
HvMYC-1	CGCTAGGAAGAAGATAGTCTCA	GGGCACTTTACCTCCAACA	qRT-PCR	
18S rRNA	CTACGTCCCTGCCCTTTGTACA	ACACTTCACCGGACCATTCAA	internal reference gene	
HvMYB-2	ATGGGGAGGATGAGG	TTATAGCGGCATGTC	Gene clone	
HvMYC-2	ATGGTGAAGCGGATC	AAAACGCCCCACTCGCT	Gene clone	
HvMYB-GFP	GGGGACAAGTTTGTACAAAAAAGCAGGC
TTCATGGGGAGGATGAGGAAGGAAG	GGGGACCACTTTGTACAAGAAAGCTGGG
TCTTATAGCGGCATGTCCACAGAG	Subcellular localization	
HvMYC-GFP	GGGGACAAGTTTGTACAAAAAAGCAGGCTTC
ATGGCGCTATCAGCTCCTCCCAGTC	GGGGACCACTTTGTACAAGAAAGCTGGG
TCCTATAGAGCTCTCTGAAGCGCTTCA	Subcellular localization	
HvMYB-YFP	CATTTACGAACGATAGTTAATTAAATGGGG
AGGATGAGGAAGGAAG	ACTGCCACCTCCTCCACTAGTTAGCGGCATGT
CCACAGAGTTT	BiFC	
HvMYC-YFP	CATTTACGAACGATAGTTAATTAAATGGCGC
TATCAGCTCCTCCCA	CACTGCCACCTCCTCCACTAGTTAGAGCTCTCT
GAAGCGCTTCA	BiFC	

The reaction system for qRT-PCR was shown in Table S3, the amplification length was 150–250 bp. Fluorescent quantitative primers were designed according Primer 5.0 software (Table 1), Primer specificity was detected by BLAST. 18S rRNA was used as an internal reference gene (Table 1), and TB Green Premix Ex Taq II (TaKaRa, Beijing, China) was used as a fluorescent dye. The Light Cycler 480 II System (Roche Diagnostics GmbH, Mannheim, Germany) was utilized to perform qRT-PCR, and the reaction system and program (Table S3) reference (Chen et al., 2023; Yao et al., 2021). All experiments were performed with three biological replicates, and the experimental data were analyzed with SPSS 19.0 for ANOVA.

Subcellular localization

Gateway technology was used to construct the expression vector. Primers with specific sites were designed for HvMYB and HvMYC genes (Table 1), and cDNA from Blue Qingke leaf was used as a template for introductory vector construction, the following experimental steps refer to previous studies (Chen et al., 2023).

HvMYB and HvMYC protein interaction assay

Using the bimolecular fluorescent complimentary technique, the interactions between the two types of transcription factors were explored in vitro. The reaction system and experimental method as previously described in Chen et al. (2023). Specifically, the specific primers of HvMYB and HvMYC genes were designed (Table 1), and the target gene was cloned using Blue Qingke leaf as cDNA template, and the target bands were recovered. The vectors used were pBiFC-YN173 and pBiFC-YC155. Vector ligation, transformation, and microscopic observation methods of the BiFC such as subcellular localization.

Results

Identification and location of HvMYB genes on chromosomes

A total of 92 Qingke HvMYB genes were identified by comparative genomics analysis and named HvMYB1–HvMYB92 based on their distribution on chromosomes. Ninety-two HvMYB genes were selectively distributed on seven Qingke chromosomes (Fig. 1). Twenty HvMYB genes were distributed on chromosome 3, which had the highest gene density (21.73%), and seven HvMYB genes were distributed on chromosome 2, which had the lowest gene density (7.60%). A total of 18, nine, 15, 10, and 13 HvMYBs were distributed on chromosomes 1, 4, 5, 6, and 7, respectively. The distribution of MYB genes on chromosomes showed that most of the MYB genes were located at the ends of the chromosomes.

Figure 1 Localization of 92 HvMYB genes on seven chromosomes.

The blue columns represent the different chromosomes, the red represents the genes distributed on the chromosomes, and the black line on the left represents the physical location (Mb) of the chromosomes, drawn with TBtools.

Physicochemical properties of HvMYB proteins

The predicted subcellular localization of the 92 HvMYB proteins showed that 82 HvMYBs were located in the nucleus, two in the mitochondria, three in the chloroplasts, and five in the cytosol, suggesting that diverse members of the same family may play roles within diverse cell groups. To further understand the properties of MYB proteins, we analyzed the physicochemical properties of all 92 HvMYB family members (Table S4). The number of amino acids ranged from 87 (HvMYB17) to 1,058 (HvMYB68). The molecular weights (MWs) ranged from 9.73 (HvMYB62) to 118.44 kDa (HvMYB68). The isoelectric points (pIs) ranged from 4.86 (HvMYB24) to 11.02 (HvMYB28). The instability index (II) ranged from 35.16 (HvMYB82) to 88.96 (HvMYB59). The aliphatic index (AI) ranged from 45.7 (HvMYB76) to 85.75 (HvMYB3). The Grand Average of Hydropathicity (GRAVY) ranged from −0.97 (HvMYB68) to −0.22 (HvMYB39). Kyte & Doolittle (1982) proposed that the higher the average hydrophilic values of proteins, the better the physicochemical properties of the overall membrane protein, while negative values indicate the soluble property of the protein. Negative GRAVY values were observed for all MYB proteins in Qingke, suggesting that MYBs are soluble proteins, which is a necessary characteristic for transcription factors (Katiyar et al., 2012).

Construction of the MYB protein phylogenetic tree and distribution of conserved domains

According to the characteristics of MYB family structural domains and transcription factors, the 92 HvMYB genes were classified into five categories, which was similar to the previous classification based on the three repeat sequences (R1, R2, and R3) of c-Myb. To further analyze the phylogenetic relationships of the HvMYB family in Qingke, we constructed a phylogenetic tree using 197 AtMYB proteins from Arabidopsis, 155 OsMYB proteins from rice, and 92 HvMYB proteins from Qingke, for a total of 444 MYB proteins (Fig. 2). Ninety-two HvMYB proteins were divided into five groups (A–E). One HvMYB was placed in Group A. Fifty-five HvMYBs were placed in Group B. Eight HvMYBs were placed in Group C. Thirteen HvMYBs were placed in Group D. Fifteen HvMYBs were placed in Group E.

Figure 2 Phylogenetic relationships of Qingke, Arabidopsis, and rice MYB proteins.

The phylogenetic tree was constructed using MEGA 7.0 with the neighbor joining (NJ) method and 1,000 bootstrap replications. All MYB domains are clustered into five branches (denoted by letters A–E): (A) MYB-related genes. (B) R2R3-MYB. (C) 1R-MYB. (D) 3R-MYB. (E) 1R-MYB.

Analysis of conserved HvMYB protein motifs and domains

To further study the homologous motif characteristics of 92 HvMYB protein members, the MEME tool was applied to analyze the motif distribution region and the frequency of the most common amino acids at each location. A total of 10 motifs (motifs 1–10) were identified in the 92 HvMYB proteins (Fig. 3; Table S5). The motifs of MYB proteins in the same group were similar in classification order, but the number of MYB protein motifs differed among diverse groups. Motif 6 was annotated as Myb_DNA-binding, which is a basic feature of MYB transcription factors. Motif 3 was found in all HvMYB proteins, except HvMYB12, HvMYB27, HvMYB76, HvMYB3, HvMYB92, and HvMYB84. Motifs 1, 2, 3, and 5 were present on the N-terminal of most HvMYB proteins. Based on the structural domain composition, the 92 HvMYB proteins were classified into 17 subfamilies, of which the most numerous was the PLN03091 superfamily domain of subfamily 2, which contained a total of 53 genes (Fig. 4; Table S4).

Figure 3 Domains and motifs in each group of the MYB proteins.

The 10 motifs predicted by MEME are shown in various colors. Dark blue, yellow, grey, blue, light blue, light grey, light green, orange, purple, and light yellow represent motifs 1–10, respectively; the diverse colors in the middle represent different domain types in the MYB family. Yellow boxes indicate CDS; blue boxes indicate UTR; and black lines indicate introns.

Figure 4 Domain composition and number of members of 17 subfamilies of 92 HvMYB proteins.

The horizontal coordinate represents the subfamily, and the vertical coordinate represents the number of members of each subfamily.

Analysis of covariance between Qingke and Arabidopsis, rice, and maize MYB families

We performed a collinearity analysis of the genomes of Hordeum vulgare (Morex_V3), Arabidopsis, Oryza sativa, and Zea mays to explore the evolutionary history of MYB genes (Fig. 5; Table S6). Only 21 copies of the Qingke MYB gene were found in the Arabidopsis genome, while 85 and 121 MYB gene pairs were found in the rice and maize genomes, respectively. The number of collinear gene pairs between Qingke and monocotyledonous plants (rice and maize) was much larger than that between Qingke and Arabidopsis, indicating that the collinearity of MYB family members between Arabidopsis and Qingke was low.

Figure 5 Analysis of covariance between barley and the MYB family of Arabidopsis, rice, and maize.

Homology analysis of MYB TFs in barley and three plants (Arabidopsis thaliana, rice, and maize). The grey line shows the co-lined blocks in the genomes of barley and the other plants, while the red line highlights the co-lined MYB pairs. There were 21 pairs of Qingke MYB gene in Arabidopsis genome, 85 pairs of Qingke MYB gene in rice genome and 121 pairs of MYB gene in the maize genome.

HvMYB gene structural diversity and main promoter cis-acting elements regulation analysis

To explore the cis-acting elements in all HvMYB gene promoters, 2 kb upstream coding sequences (CDSs) of each gene were extracted and searched in the PlantCARE database. The cis-acting elements of the 92 HvMYB genes promoter regions were analyzed (Fig. 6; Table S7), the results showed that the promoter regions of 92 genes contained 42 cis-acting elements associated with stress and hormones (Table S8). Among them are abscisic acid responsive, auxin, and gibberellin, which are related to plant physiological regulation, light regulation, and response to biological stress. Abiotic stress and immune response elements, such as MYB-binding sites, low temperature, methyl jasmonate, salicylic acid, defense, and stress response, were also included. Most HvMYB genes contained ABA-responsive, light-responsive, and MeJA-responsive elements. HvMYB genes could be associated with plant physiology, biotic and abiotic stress, and immune responses. HvMYB genes have been speculated to be inducible promoters. The presence of various cis-acting elements in gene promoters indicates that these genes have diverse functions.

Figure 6 Prediction and exon–intron structure of cis-acting elements in the promoter regions of HvMYB genes.

The MYB protein contains a total of 42 cis-acting elements, and different colors represent different element names.

Screening of the HvMYB genes related to anthocyanin synthesis

To further understand the expression levels of 92 HvMYB genes in different Qingke varieties, RNA-seq was performed on ‘White 91-97-3’ and ‘Blue Qingke’ at the milk, soft dough, and mature stages. White 91-97-3 and Blue Qingke contained 10,940 differential genes. The 92 HvMYB genes intersected with the differential genes of the two varieties, which contained 41 differential genes (Fig. 7). The differential genes of both Blue Qingke and White Qingke have similar expression patterns, that is, the expression level was higher in the mature stage, while the expression level was lower in the milk stage. It is inferred that genes regulating Qingke blue grain traits mainly play a significant role in the middle stage of grain development (Fig. 8; Table S9). In view of the location interval of blue grain Qingke in this experiment, HvMYB53 was identified as the key gene controlling blue grain Qingke, and HvMYC interacting with HvMYB53 was screened in RNA-seq, these two genes were selected as key genes for subsequent analysis. Subsequently, we performed qRT-PCR analysis of these two genes, the primer sequence was shown in Table 1. RNA-seq and qRT-PCR (Table S10) analysis showed that the two genes had similar expression patterns: that is, the two genes were highly expressed in the soft dough stage (Figs. 9A and 9B).

Figure 7 Venn diagram of Blue Qingke and White Qingke RNA-seq differential gene and 92 identified genes.

Figure 8 Expression profiles of 41 differential genes in Blue Qingke and White Qingke.

Blue represents lower expression and red represents higher expression. White1, White2, White3, Blue1, Blue2, and Blue3 represent the milk, soft dough, and mature stages of White 91-97-3 and Blue Qingke, respectively. FPKM values of HvMYB genes transformed by log2 and heatmap constructed by TBtools. The numbers in the rectangles indicate the expression in RNA-seq.

Figure 9 Relative expression of HvMYB53 and HvMYC in White 91-97-3 and Blue Qingke.

The 18S rRNA were used as internal. W1, W2, and W3 represent the milk, soft dough, and mature stages of White 91-97-3, respectively. B1, B2, and B3 represent the milk, soft dough, and mature stages of Blue Qingke, respectively. The error line represents the standard deviation (n = 3). Different capital letters indicate significant differences between the two varieties in the three periods (P < 0.01). (A) The relative expression of HvMYB53. (B) The relative expression of HvMYC.

Subcellular localization and BiFC analysis of HvMYB and HvMYC

We found that HvMYB and HvMYC proteins interact, based on the results of protein-protein interaction (PPI) of RNA-seq in our laboratory (Table S11). To understand the protein characteristics of HvMYB and HvMYC, the subcellular localization of this proteins were detected. Nicotiana benthamiana leaves expressing the green fluorescent protein (GFP) were analyzed using a confocal microscope. HvMYB-GFP was expressed in the nucleus and cell membrane, HvMYC-GFP was expressed in the nucleus, cell membrane, and cytoplasm. Therefore, HvMYB was localized in the nucleus and cell membrane, HvMYC was localized in the nucleus, cell membrane, and cytoplasm (Fig. 10). HvMYB and HvMYC fluoresce yellow in the nucleus and cell membrane, indicating an interaction (Fig. 11).

Figure 10 Subcellular locations of HvMYB and HvMYC.

HvMYB-GFP was located in the nucleus and cell membrane, HvMYC-GFP was located in the nucleus, cell membrane, and cytoplasm (green). Images of GFP, chlorophyll autofluorescence, bright field, and GFP merged with bright field (Merge) are shown. The scale bars are 100 μm.

Figure 11 Validation of interaction between HvMYB and HvMYC were analyzed via BiFC.

It was found that HvMYB and HvMYC interact to form biomolecular fluorescent complexes (yellow). Images of YFP, chlorophyll autofluorescence, bright field, and YFP merged with bright field (Merge) are shown. The scale bar indicates 50, 100, and 200 μm.

Discussion

The MYB gene family is associated with a variety of plant-specific cellular functions (Shen et al., 2013). The main feature of the MYB protein is the localization of three regular tryptophan (W) residues with each DBD, and the abundance of the MYB gene in a species may be related to genome duplication (segmentation/tandem) rather than genome size (Zheng et al., 2006). MYB genes are characterized as transcription factors when they contain at least two MYB repeats (R) and explicitly recognize DNA motifs to regulate gene transcription (Yu et al., 2017). MYB transcription factors play an extensive role in plant growth and development, hormone signaling, and coping with stress responses. The gene has been studied more in Arabidopsis (Kirik et al., 1998), rice (Katiyar et al., 2012; Mullick et al., 1958), and maize (Du et al., 2012) and less in Qingke. Transcription factors are usually co-expressed with their downstream structural genes, and similar expression patterns are often an effective way to identify transcription factors that regulate secondary metabolism (Yu et al., 2021). Therefore, studying the Qingke MYB gene family and comparing it with its homologous plant genes will help explain the evolution of this gene family during the natural formation of Qingke.

The 92 HvMYB genes in this study were named HvMYB1–HvMYB92 based on their location on the chromosome, which distinguishes them from previous studies that named them based on gene annotation and structural domain type. Analysis of motifs and intron–exon patterns of full-length protein sequences helps identify various domains and further confirms the results of phylogenetic analysis (Giacomelli et al., 2010; Wang et al., 2011). The MYB family members in the same group have a similar structure, with all MYB proteins except HvMYB12, HvMYB27, HvMYB76, HvMYB3, HvMYB92, and HvMYB84 containing motif 3 (pink) and most MYB proteins containing motifs 1 (dark blue) and 2 (yellow). Motifs 1, 2, 3, and 5 appeared sequentially at the N-terminus of HvMYB proteins, which is consistent with the findings of Yu et al. (2021). In addition, motif 5 was distributed before motif 3, and motif 7 occurred before motif 1 in most HvMYB proteins, suggesting that HvMYB proteins are highly conserved and that motif differences among HvMYB proteins may be related to the biological functions of their specific genes. Exon–intron structure diversity and cleavage pattern significantly affected the number and expansion of gene families (Wei et al., 2016). According to rice studies, intron loss occurs faster than intron addition after fragment replication (Lin et al., 2006). Our study found that the number of introns was proportional to the length of the gene. In Qingke, HvMYB75, HvMYB59, HvMYB72, HvMYB91, HvMYB15, HvMYB16, HvMYB18, HvMYB18, HvMYB62, HvMYB63, and HvMYB8 had no introns, indicating that these MYB branches evolved later than the other groups.

The highly conserved tryptophan (W) residues distributed in the third helix are significant for the DNA-binding activity of MYB proteins, which indicates functional conservation between different plant species, but sequence conservation between plants does not necessarily imply functional conservation (Du et al., 2012). In this study, 92 HvMYB genes were classified into five categories, the resulting functional conservation between the Qingke HvMYB genes and their homologs suggests the origin and evolutionary diversity of plant MYB genes; species-specific groups/subgroups may evolve or be lost during evolution, leading to functional divergence (Du et al., 2012). Transcription factors specifically bind to cis-acting elements to regulate the expression of related target genes, and multiple cis-acting sites determine the diversity of the regulatory functions of transcription factors (Lei et al., 2021; Ning et al., 2017; Zhao et al., 2021). Further analysis of the cis-acting elements of diverse genes showed that HvMYB genes are presumably associated with plant physiology, biotic and abiotic stress, and immune responses, and it has been hypothesized that HvMYB genes are inducible promoters and that the presence of various cis-acting elements in the promoters of the genes may imply that these genes have diverse functions.

Loss of a structural domain does not always lead to negative consequences, and although gene function is temporarily lost or altered, it may lead to the re-expansion of the gene family (Liu et al., 2022). The major structural domains of MYB proteins are PLN03091 and PLN03212, which is consistent with the findings of Yu et al. (2021). F-box proteins are involved in various life activities, such as cell cycle regulation, apoptosis, and signal transduction, and they play vital roles in maintaining normal plant growth and development and mediating abiotic stress responses (Xu et al., 2023). The SANT structural domain, which is found mainly in proteins involved in chromatin function and usually recognizes histone tails, is a novel motif found in numerous eukaryotic transcriptional regulatory proteins, and it shares homology with the DNA-binding structural domain of c-MYB, although it is unlikely to bind DNA (Boyer et al., 2002; Grüne et al., 2003). Arabidopsis AtMYB96 is an R2R3-type MYB transcription factor that regulates the drought stress response by integrating ABA and auxin signals, and genes in the same branch as this gene may have the same function (Joon et al., 2009). Some of the structural domains in MYBs contain highly conserved SHAQKY (F/Y) motifs in the third predicted α-helix, and the 1R structural domains of OsMYBs and StMYB1 are highly conserved, suggesting that other amino acid residues upstream of the SHAQK (Y/F) F motif and within the predicted third α-helix may contribute to the recognition of specific DNA sequence elements (Lu et al., 2002). The above structural domains are present in only a few MYB proteins, indicating that they are not the major structural domains of MYB proteins. Gene collinearity analysis among diverse species showed that Qingke and maize had the most collinear gene pairs, followed by rice and Arabidopsis; this difference may be related to Qingke’s genome characteristics and evolution.

Garcia-Gimenez et al. (2022) hypothesized that MYB transcription factors may positively regulate the accumulation of (1,3; 1,4)-β-glucan in barley primary cell walls. Recent studies have also described novel roles for MYBs as repressors, negatively affecting traits such as secondary cell wall biosynthesis and cold tolerance (Jende-Strid, 1993; Mullick et al., 1958). Previous studies have located the HvMYB4H gene in blue barley on chromosome 4H (Jia et al., 2020), indicating that the MYB gene is a significant factor in the blue color of wheat crops. Subcellular localization predicts that most HvMYB genes are located primarily in the nucleus, and it is hypothesized that MYB genes perform important functions primarily in the nucleus. The flavonoid biosynthetic pathway is transcriptionally regulated by the MBW complex, whose activation or repression is mainly determined by MYB transcription factors through binding to structural gene promoters and common bHLH and WD40 factors (Kyte & Doolittle, 1982). In potato, when three MYB genes (StAN1, StMYBA1, and StMYB113) are co-expressed with the bHLH gene, anthocyanin synthesis in tobacco can be regulated (Liu et al., 2016). The ZmPAC1 gene in maize can regulate anthocyanin synthesis when co-expressed with ZmR1 (bHLH) and ZmC1 (MYB) (Carey et al., 2004). These results suggest that the combination of MYB and bHLH to regulate anthocyanin synthesis. BiFC results showed that HvMYB and HvMYC can form protein interactions. We speculate that the joint role of HvMYB and HvMYC in the synthesis of anthocyanins in Qingke seed colour may be stronger than that of a single transcription factor. Expression analysis of the differential genes and two reciprocal genes showed that all the genes had higher expression almost at the soft dough stage and the mature stage, and lower expression at the milk stage. It is hypothesised that the genes regulating Blue Qingke mainly start to function in the middle stage of seed development, which may also be related to their involvement in different biological processes or functions.

Conclusion

In this study, we identified 92 Qingke HvMYB genes using protein-motif interactions. Among them, HvMYC protein interacting with HvMYB53 was found in Blue Qingke and White 91-97-3 RNA-seq data, which was also proved by BiFC method. HvMYB53 and HvMYC genes play a significant role in Qingke grain coloring, and their expression patterns are significant differences among different varieties. This study not only proved the key role of HvMYB53 and HvMYC genes in anthocyanin biosynthesis, but also provided a theoretical basis for further research on the regulatory mechanism and functional evolution of Qingke anthocyanin biosynthesis.

Supplemental Information

Supplemental Information 1 Sequence of HvMYB53 and HvMYC.

Supplemental Information 2 RNAseq power.

Supplemental Information 3 cDNA reverse transcription reaction system.

Supplemental Information 4 qRT-PCR reaction system.

Supplemental Information 5 Physicochemical properties and domain composition of 92 HvMYB proteins.

Supplemental Information 6 Raw data of protein domains.

Supplemental Information 7 Collinearity of barley with rice, Arabidopsis, and maize.

Supplemental Information 8 Raw data for cis acting elements.

Supplemental Information 9 Analysis of promoter region elements of 92 HvMYB genes.

Supplemental Information 10 Raw RNA-seq data of 41 differential genes from blue and white Qingke.

Supplemental Information 11 Raw data from HvMYB53 and HvMYC RNA-seq and qRT-PCR.

Supplemental Information 12 PPI results of interacting proteins HvMYB and HvMYC in blue and white Qingke RNA-seq.

Supplemental Information 13 MIQE checklist.

Supplemental Information 14 Protein sequences.

Additional Information and Declarations

Competing Interests

Author Contributions

Data Availability

The authors declare that they have no competing interest.

Hongyan Li conceived and designed the experiments, performed the experiments, analyzed the data, prepared figures and/or tables, authored or reviewed drafts of the article, and approved the final draft.

Youhua Yao analyzed the data, prepared figures and/or tables, and approved the final draft.

Xin Li analyzed the data, prepared figures and/or tables, and approved the final draft.

Yongmei Cui analyzed the data, prepared figures and/or tables, and approved the final draft.

Likun An analyzed the data, prepared figures and/or tables, and approved the final draft.

Baojun Ding analyzed the data, prepared figures and/or tables, and approved the final draft.

Xiaohua Yao conceived and designed the experiments, analyzed the data, prepared figures and/or tables, authored or reviewed drafts of the article, and approved the final draft.

Kunlun Wu conceived and designed the experiments, analyzed the data, prepared figures and/or tables, authored or reviewed drafts of the article, and approved the final draft.

The following information was supplied regarding data availability:

The MYB family hmm data is available at the Pfam database: https://www.ebi.ac.uk/interpro/entry/pfam/PF00249/.

The FASTA and GTF data for the barley genome is available at the EnsemblPlants database: e!DAL-Plant Genomics & Phenomics Research Data Repository, DOI 10.5447/ipk/2021/3.

The Arabidopsis genome data is available at: https://www.arabidopsis.org. The Arabidopsis MYB family classification is available at The Arabidopsis Information Resource: https://www.arabidopsis.org/locus?key=127904.

The rice genome data is available at https://rice.uga.edu.

The maize genome data is available at: https://db.cngb.org/zeamap.

The structural domain analysis data is available at SMART (http://smart.embl-heidelberg.de) and NCBI (https://www.ncbi.nlm.nih.gov/Structure/bwrpsb/bwrpsb.cgi).

The physicochemical properties of proteins data is available at the Expasy Protparma: http://web.expasy.org/protparam.

The subcellular localization data is available at Zenodo: https://doi.org/10.5281/zenodo.14010682.

The RNA-seq data of Qingke with different grain colors were obtained from the preliminary test in our previous studies are available at NCBI SRA: SRR29701474, SRR29701475, SRR29701476, SRR29701477, SRR29701478, SRR29701479, SRR29701480, SRR29701481, SRR29701482, SRR29701483, SRR29701484, SRR29701485, SRR29701486, SRR29701487, SRR29701488, SRR29701489, SRR29701490, SRR29701491. We retrieved the number of enzyme fragments per kilobase million (FPKM) values for the MYB family at each developmental stage from RNA-seq.

The structure required domain of the original data and the raw data of cis-acting elements, qRT-PCR, and RNA-seq are available in the Supplemental Files.

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
