# Peer review of "Comparative genomics analysis of the MYB gene family in barley: preliminary insights into evolution and biological function in Blue Qingke"

_PeerJ, doi:10.7717/peerj.18443_

## Round 0.1 · original submission · Major Revisions

Dear Dr. Li,

Your manuscript has been evaluated by two experts. Both believe your work is important, but they raised issues regarding data presentation and methodogy.

See for example "Reviewer one analysis - the methods section needs heavy rewriting to be ready for publication.Finally, the current experimental design failed to use and compare itself to previous work. Barley MYB genes were already listed in the Plant transcription factor database".

Reviewer 2 provides a similar opinion

"The manuscript requires significant revision. Follow proper writing format in the different sections: MM, results, and discussions. Avoid repeating MM in results and discussions; results in discussion. Focus on presenting your results.
Avoid detailed descriptions of standard techniques, particularly in results and discussion. Where necessary, provide short descriptions in MM".

Thus, it is important that the manucript undergoes extensive rewriting and a point-by-point reply to each reviewer is provided in the revised version of the manuscript.

Reviewer 1 ·

Basic reporting

Li et al. paper report a genome-wide study of MYB genes in barley, with an application to hulless barley (Qingke). The study is well carried out but suffers from insufficient details in the paper and some necessary analyses are missing.
The paper is written in clear English, apart from a few words where the translation was incorrect (for example l.72: “endemic” is not used for genes, and it is “biotic” rather than “biological”. The introduction is sometimes messy, as exemplified by paragraph l65-91 which is a juxtaposition of facts without clear coherence. However, the main elements are present and the introduction needs only some light revision.
The methods section cites websites where proper references for the tools are needed, in particular in light of the life expectancy of websites.
Figures are often unreadable, like 2 (font too small, it is not possible to see where are the barley genes), 3, and 5. For those 2, in addition to the font problem, there is too much data, making it hard to understand. A choice of what to represent must be made. Similarly, the wrong choice was made for Figure 4. The information provided is not useful. We can see the colinearity between species (not conserved, unsurprisingly), but we can’t see how many genes of each genome have homologs on the other one and how many don’t, which is, to my opinion, the important piece of information. Table 2 might have been useful in a paper with numerous abbreviations, but It’s not the case here. It would be much more important to disambiguate all abbreviations the first time they appear (not the case currently) than to provide this table. In addition, legends of all figures and tables are often too short to understand them properly.
Concerning raw data, RNAseq data are missing and should be submitted to an archive.
More importantly, I’m quite concerned by the way the results are presented. All over the paper, it is referred to Qingke MYB genes, while the analyses were done on the barley reference genome, which is of the Morex cultivar, not hulless. This simple, easy-to-correct, mistake, compromises the whole paper. Even the title is properly misleading.

Experimental design

The research fits with the scope of the journal. The research question is clear and fills a knowledge gap.
However, the methods section is insufficient. It is currently impossible to repeat the work with the provided information. In addition to the missing references already stated previously, the used parameters of the softwares are never provided. Some methods are completely missing: how was the RNAseq data processed? What are the PCR cycles? The mixes (provided in supplementary but should be in the text body)? How were plants grown? The cloning protocol is also not detailed enough nor is the vector used for BiFC. To conclude here: the methods section needs heavy rewriting to be ready for publication.
Finally, the current experimental design failed to use and compare itself to previous work. Barley MYB genes were already listed in the Plant transcription factor database. This is based on an older version of Morex reference genome but the comparison between the 92 genes obtained in this paper and the 99 previously listed is mandatory. Also, previous papers have identified MYB genes and even named them. It is also mandatory to compare their nomenclature with the one in this paper.

Validity of the findings

The data presented seems sound, but I don’t have enough data to ensure it:
L239-246: I don’t have enough data to know if you found homologs for all rice genes, nor, more generally, to know if you didn’t miss some genes in barley.
l271-281, no statistic is provided on the RNAseq data. Expression is checked in blue and white quingke, but at no point, sequences are checked there, so we don’t even know if the primers are good enough for those barley varieties. 41 DEG are cited, but DEG compared to what? The information is missing. HvMYB53 is described as the key gene but I didn’t see any supporting data for that. Why? Is it the previously identified HvAnt1/HvMpc2?
L.284: where can I see the PPI data?
l.222-238 and l.252-263 are catalogs of data that would be better represented as figures or supplementary figures. For the 1st paragraph, there is no table in supplementary showing the classification of each gene. It could be added to S5. For the second paragraph, a table a bit simpler than S7 would be nice, where for each gene, domains present are presented in a double-entry table.

Additional comments

To conclude, this paper closes a knowledge gap by studying a family of transcription factors in barley that has not been studied in detail. However, the title and the paper are misleading, as the study was not carried out in blue quingke but in barley reference genome. The methods section is largely incomplete, some raw data is missing and the comparison with the previous knowledge is missing. The paper need heavy revision before being ready for publication

Reviewer 2 ·

Basic reporting

Comparative genomics analysis of the MYB gene family in Blue Qingke: Preliminary Insights into Evolution and Biological Function reports the studies on MYB using RNA sequencing, quantitative reverse transcription polymerase chain reaction, and bioinformatic method. This work appears to be the first Genome-wide transcriptome analysis for the Qingke species.
General comments
The manuscript requires significant revision.
Follow proper writing format in the different sections: MM, results, and discussions.
Avoid repeating MM in results and discussions; results in discussion
Focus on presenting your results.
Avoid detailed descriptions of standard techniques, particularly in results and discussion. Where necessary, provide short descriptions in MM.

Experimental design

Lines 25-26: This study used comparative genomics methods to analyze gene and protein structure, protein physicochemical properties, chromosome localization, and evolution.
Is the method section, not the result?

Line 29-30: “At the milk stage, soft dough stage, and mature stage, white and 30 blue Qingke grains were selected for RNA-seq, among which two genes interacted.” It should also be in the material and method section.

Validity of the findings

Line 92-99: “Currently, studies on the MYB gene family have focused on Arabidopsis, rice, and maize, while 93 fewer studies have been conducted on the Qingke MYB family. Based on published reports of MYB genes 94 in Arabidopsis thaliana and rice, 92 Qingke HvMYB genes were obtained by genome-wide analysis. 95 Chromosome localization, gene and protein structure, protein physicochemical properties, conserved 96 domains, protein conserved motifs, promoter cis-acting elements, phylogenetic relationships, collinearity 97 with Arabidopsis thaliana, rice, and maize were analyzed using bioinformatics methods. These results 98 laid the foundation for the research and application of MYB transcription factors in the grain color of 99 cereal crops.”
This part needs to be referenced. Which paper published the GWAS on the MYB family? Who worked on the gene structure? As you mentioned, this work is the first for Ginke; it contradicts your point of introduction. Who researched the 96 conserved protein motives? Give reference.

Result section

Line 291-293: “Using the bimolecular fluorescent complementary technique, the interactions between the two types 292 of transcription factors were explored in vitro”
: Move it to the material and method section

Discussion section
Line 310-311: “In this study, ninety-two HvMYB genes were identified in Qingke based on homology matching and conserved structural domains.” Move it to material and method.

Additional comments

Don't repeat results in the discussion section as presented in the result section. Focus on justifying (why you have those results) your results in line with previous studies and/or your observations. For example, lines 240 to 246

Start sentences with a discussion style
E.g., 'The MYB gene family is associated with,

---

## Round 0.2 · Minor Revisions

Dear Dr. Li,

Although the reviewer finds your manuscript has largely improved from the previous version, he requires and I agree that you should either revise the threshold or argue why you did not use the standard threshold for your analysis.

"The methodology section discusses the statistical power of the RNA-seq data as 0.37 (Page 11), which may be considered insufficient for robust conclusions. You might consider using a higher threshold for greater reliability. As Smith et al. (Journal of Genetics, 2019) suggest, a power above 0.80 is generally required for conclusive genetic studies".

After you revise your manuscript, please send the manuscript back to our evaluation

Reviewer 2 ·

Basic reporting

The manuscript titled "Comparative genomics analysis of the MYB gene family in Blue Qingke: Preliminary Insights into Evolution and Biological Function, Here is the comment to the author.
The research provides a solid foundation by leveraging bioinformatics methods and RNA-seq to explore the MYB gene family in Qingke, an underrepresented species.

The manuscript offers valuable insights into the anthocyanin biosynthesis pathway, with implications for both functional genomics and plant breeding.

Introduction Detail (Lines 39-99)
The introduction would benefit from a deeper discussion of how the study contributes to filling a specific gap in MYB transcription factor research, especially focusing on under-researched Qingke variants. Consider expanding the description of the knowledge gap filled by this work, particularly in the context of anthocyanin biosynthesis.

Although the manuscript uses clear, professional English grammatical errors, such as in lines 100-110, disrupt the flow of the text.

Experimental design

The methodology section discusses the statistical power of the RNA-seq data as 0.37 (Page 11), which may be considered insufficient for robust conclusions. You might consider using a higher threshold for greater reliability. As Smith et al. (Journal of Genetics, 2019) suggest, a power above 0.80 is generally required for conclusive genetic studies.

Validity of the findings

The conclusions are well stated and linked to the original research.

Additional comments

The MYB family...studies of Qingke are lacking." This sentence could be rephrased to avoid awkward phrasing. Consider: "The MYB family...but studies specifically on Qingke remain limited."

---

## Round 0.3 · accepted · Accept

Congratulations on the acceptance of your manuscript

Reviewer 2 ·

Basic reporting

Clear and unambiguious.

Experimental design

Original primary research.

Validity of the findings

Conclusions are well stated.

Additional comments

None.